# Bioengineered Skin from a Platelet-Derived Hydrogel Repairs Full Thickness Wounds in a Pre-Clinical Mouse Model

**DOI:** 10.3390/ijms26209988

**Published:** 2025-10-14

**Authors:** Md. M. Rahman, Carlos L. Arellano, Ilia Banakh, Denese C. Marks, Irena Carmichael, Frank Arfuso, Cheng Hean Lo, Heather Cleland, Shiva Akbarzadeh

**Affiliations:** 1Skin Bioengineering Laboratory, Victorian Adult Burns Service, Alfred Health, 89 Commercial Road, Melbourne, VIC 3004, Australia; mostafiz.rahman@monash.edu (M.M.R.); carlos.arellano1@monash.edu (C.L.A.); ilia.banakh@monash.edu (I.B.); cheng.lo@monash.edu (C.H.L.); h.cleland@alfred.org.au (H.C.); 2Department of Surgery, Monash University, 99 Commercial Road, Melbourne, VIC 3004, Australia; 3Australian Red Cross Lifeblood, 17 O’Riordan Street, Alexandria, NSW 2015, Australia; dmarks@redcrossblood.org.au; 4MMI, Monash University, 99 Commercial Road, Melbourne, VIC 3004, Australia; iska.carmichael@monash.edu; 5School of Human Sciences, The University of Western Australia, 35 Stirling Highway, Perth, WA 6009, Australia; frank.arfuso@uwa.edu.au

**Keywords:** platelet-derived biological material, wound repair, engineered skin, basal keratinocytes

## Abstract

Despite advancement in skin engineering, native skin grafting remains the gold standard in clinical settings. We have previously demonstrated that a platelet-derived hydrogel (PG) can act as a scaffold to engineer a semi-mature bilaminar human skin equivalent (PG-HSE). In this study, PG-HSE was grafted on full thickness wounds in athymic mice. PG-HSE was compared with native skin autografts and a clinically proven bilaminar skin graft that utilises a single layer NovoSorb^®^ polyurethane biodegradable temporising matrix (plus plasma) as the scaffold (BTM-HSE). The graft analysis revealed PG-HSE-grafted wounds were fully epidermised in two weeks and the level of inflammatory markers, *CXCl1*, *CXCl2*, *IL1β*, and *IL-6* transcripts, in grafts were at similar levels to their levels in autografts. This coincided with higher expression of *COL1A2*, *COL3A1*, and *COL5A1* transcripts in PG-HSE grafts, compared to autografts and BTM-HSE grafts. Moreover, a higher deposition of both Col I and Col III was detected in the PG-HSE graft wound bed, when compared to the BTM-HSE graft wound bed. Conversely, BTM-HSE grafts showed a higher level of integrins, *ITGA2*, *ITGA3*, *ITGA5*, *ITGA6*, *ITGAV*, and *ITGB1*, at the RNA level, suggesting a stronger cell–scaffold interaction. In summary, we have shown although both PG and single layer BTM foam (plus plasma) are effective scaffolds for skin engineering, some key aspects of wound repair, including a reduction in inflammation and an increase in collagen deposition, are achieved with the platelet-derived hydrogel. The long-term effect of these scaffolds on wound scarring remains to be investigated.

## 1. Introduction

Wound repair remains a clinical challenge in both acute (e.g., major burns) and chronic (e.g., diabetic foot ulcer) scenarios, where spontaneous healing is not achievable in a timely fashion. Biomaterial scaffolds (with seals) have revolutionised clinical care in such scenarios. Scaffolds aim to provide an extracellular matrix (ECM) to enhance cell infiltration and form neo-dermis, thereby reducing wound contraction. The seal provides a temporary wound closure, which inhibits wound infection [1,2]. However, establishing a permanent protective barrier can only be achieved by grafting patients’ own epidermis. Apart from the traditional autologous thin skin graft that is harvested from a donor site, engineered skin from expanded basal keratinocytes can provide a viable alternative [3].

In order to provide an appropriate microenvironment for basal keratinocytes, a wide range of biomaterials such as collagen-based matrices, hydrogels, silk, biodegradable polyurethane, or their combinations have been tested as scaffolds over the last few decades, each with inherent advantages and disadvantages [4]. We and others have previously shown plasma, alone or in combination with single layer (without a seal) NovoSorb^®^ polyurethane biodegradable temporising matrix, BTM foam (PolyNovo Biomaterials Pty Ltd., Port Melbourne, VIC, Australia), to be an effective scaffold for neo-epidermis engineering in vitro and in vivo [5,6,7,8,9,10]. Single layer BTM foam is fully synthetic and biocompatible. However, when used alone, it does not allow normal upwards basal keratinocyte maturation and stratification due to its large pore size. It also has a slow degradation rate in patients [11,12]. Plasma gelation into single layer BTM foam has been reported to allow neo-dermis/neo-epidermis formation that closed large wounds in a patient [13]. Fibrinogen, in plasma, can block the pores in single layer BTM foam and create a smooth surface for epidermisation. Fibrinogen can also assemble within matrix fibrils of fibroblasts or epithelial cells, enhancing wound closure by increasing both cell proliferation and migration [14,15]. Fibroblasts bind to fibrinogen via cell surface integrin α5β1 [16].

Although human plasma is an effective scaffold, its availability in large volumes would present challenges in translation to clinics. Worldwide demand for plasma is high, as it is collected for fractionation as well as for transfusion. Alternatively, platelets can be used to form a hydrogel scaffold. Platelets are generally in less demand compared to plasma, due to their limited clinical use and shorter shelf life (seven days), versus plasma (up to a year).

To address these issues, we have previously developed a platelet-derived hydrogel (PG) that is rich in pro-epidermisation growth factors, such as insulin growth factor (IGF) and epidermal growth factor (EGF). It also stimulates collagen deposition in a full thickness wound [17]. In order to construct a composite neo-dermis/neo-epidermis skin, primary adult fibroblasts were added to a PG prior to seeding basal keratinocytes to engineer a platelet-derived human skin equivalent (PG-HSE) in vitro [18]. In PG-HSE, fibroblast attachment results in hydrogel retraction in a similar way to retraction of a nucleated blood clot [16]. Fibroblasts also play a crucial role in basal keratinocyte attachment and growth by secreting ECM and increasing hydrogel stiffness, which are crucial for cell attachment and growth [19]. PG-HSE showed a near native balance of keratinocyte proliferation and differentiation, forming a mature skin in vitro [18].

One of the key factors in clinical translation of any engineered skin is its ability to vascularise in a timely fashion. We have previously shown both acellular PG and acellular single layer BTM foam enhance vascularisation in full thickness wounds in mice [10,17]. The objective in this study was to assess PG-HSE graft on wound repair, compared to BTM-HSE graft and the traditional native skin autograft. Particularly, the effect on wound inflammation, collagen deposition, and wound closure was characterised.

## 2. Results

### 2.1. Platelet-Derived Hydrogel Is an Effective Scaffold for Constructing a PG-HSE Graft and Closing Wounds in a Mouse Model

As the initial step in constructing PG-HSE, we showed that the platelet-derived precipitate was non-toxic to adult dermal fibroblasts. In fact, the platelet-derived precipitate stimulated fibroblast proliferation in a similar fashion to foetal bovine serum (Appendix A). The PG scaffold was constructed by gelation of platelet-derived precipitate and was then populated with dermal fibroblasts and basal keratinocytes. The construct was allowed to mature for 5 days and form the PG-HSE (Appendix A). As a clinical comparative, the BTM-HSE was constructed by populating the single layer BTM foam (plus human plasma) scaffold with dermal fibroblasts and basal keratinocytes, and the skin was matured for 12 days.

Full thickness wounds were splinted on the dorsal side of athymic nude mice. They were grafted with PG-HSE, BTM-HSE, or a native full thickness autograft. A cohort of mice was left without grafting. Figure 1A presents the macroscopic images of wound closure. Wound size was measured on days 5, 10, and 14 post-grafting (Appendix A). Grafts were harvested on day 14 for histological and molecular analysis. Data showed that all grafted and ungrafted wounds maintained their size without any significant contraction for at least 10 days. Thus, splinting effectively slowed down wound contraction. On day 14, the ungrafted wounds were significantly contracted compared to day 0 (*p* ≤ 0.001), indicating the effectiveness of skin grafting in reducing wound contraction. Wounds grafted with PG-HSE contracted, but to a lesser degree, 14 days post-grafting (*p* ≤ 0.05). Yet, wounds grafted with either autograft or BTM-HSE maintained their size throughout the experiment.

To show wound epidermisation was due to grafted human cells and not host cell migration or wound contraction, grafts were analysed by immunofluorescence using an anti-human involucrin antibody. This antibody marks differentiating keratinocytes and does not cross react with mouse involucrin (Figure 1B,C). In PG-HSE grafts, involucrin was present in the suprabasal layer in a similar fashion to native adult human skin (albeit a thicker layer). In BTM-HSE grafts, however, the differentiation pattern was perturbed both before and after grafting. The differentiating human adult keratinocytes were detected within the scaffold as well as the upper epidermis. This indicated that despite attempts to block pores in the scaffold by prior plasma gelation, keratinocytes still migrated into single layer BTM foam pores. As expected, human-specific involucrin was absent in autografts. Whole grafts were analysed by immunofluorescence using a human-specific antibody against the nuclear marker Ku80 that does not cross react with the mouse protein. Ku80 was used as an independent marker, confirming the persistence of human adult fibroblasts and basal keratinocytes in PG-HSE and BTM-HSE grafts two weeks post-grafting (Figure 1D). Some non-specific human Ku80 was detected in autografts and ungrafted wounds (*p* ≤ 0.05).

### 2.2. PG-HSE Grafts Show Lower Transcription Levels of Inflammatory Markers, Compared to BTM-HSE Grafts

For wound repair to progress, inflammation is required to subside. The inflammation level in the engineered skin graft microenvironment was measured by real-time RT-PCR. Members of the chemokine C-X-C ligand, *CXCL-1* (*p* ≤ 0.0001) and *CXCL-2* (*p* ≤ 0.0001), as well as pro-inflammatory interleukins, *IL-1β* (*p* ≤ 0.01) *and IL-6* (*p* ≤ 0.001), were significantly lower in PG-HSE grafts and autografts compared to BTM-HSE grafts (Figure 2A). Similarly, a key inflammation master regulator, prostaglandin-endoperoxide synthase 2, *PTGS2* (that encodes *COX-2*) was low in PG-HSE grafts and autografts (*p* ≤ 0.0001) (Figure 2B). Overall, the data suggest PG-HSE grafts, similar to autografts, have a lower inflammation status compared to BTM-HSE, two weeks post-grafting.

### 2.3. Basal Keratinocytes Are Sustained in Both PG-HSE and BTM-HSE Grafts

The persistence of keratinocyte progenitors in grafts is vital for HSE survival. Several cell adhesion molecules including integrins were used as markers to detect basal keratinocytes (Figure 2). Here, we confirmed wound healing by haematoxylin and eosin staining (Figure 3A) and detected basal keratinocytes in both PG-HSE and BTM-HSE grafts using CD29 (Integrin β1) marker at the transcription (Figure 2C) and protein level using an immunofluorescence assay (Figure 3C). CD29^+^ adult basal keratinocytes were selectively expanded more in the neo-epidermis in BTM-HSE grafts (*p* ≤ 0.01), compared to PG-HSE grafts. Similarly, other integrin subunits, including integrin subunit alpha *ITGA2* (*p* ≤ 0.0001) and *ITGA3* (*p* ≤ 0.0001), that normally heterodimerise with subunit beta (*ITGB1*) on basal keratinocytes, were significantly overexpressed in BTM-HSE grafts. Although *ITGAV* was upregulated in BTM-HSE grafts (*p* ≤ 0.0001), its integrin subunit beta 5 (*ITGB5*) heterodimer expression was significantly lower (*p* ≤ 0.0001). This analysis, however, is not able to distinguish between the human integrins in grafts and their mouse counterpart due to their high homology, resulting in primer cross-reactivity with the host sequence.

Cadherin 1 or E-cadherin (*CDH1*), which controls stratification, is one of the two most abundant cadherins expressed in the epidermis (Figure 2). Similarly to the upregulation of basal keratinocytes integrin markers, *CDH1* is upregulated in BTM-HSE grafts (*p* ≤ 0.0001). Epidermal growth factor receptor (*EGFR)* and heparin-binding epidermal growth factor-like growth factor (*HB-EGF*) expression was also upregulated in BTM-HSE grafts, compared to PG-HSE (*p* ≤ 0.0001). Additionally, expression of *TGF-β1* and its downstream signalling molecule, signal transducer, and activator of transcription 3 (*STAT3*) was upregulated in BTM-HSE grafts, compared to PG-HSE (*p* ≤ 0.0001). Overall, although both PG-HSE and BTM-HSE grafts showed a significant overexpression of pro-epidermisation molecular markers, compared to autografts, their expression levels were higher in BTM-HSE, compared to PG-HSE.

The canonical Wnt pathway is known to control spontaneous wound repair [20]. Evidence of Wnt canonical pathway activation in PG-HSE grafts was observed since the expression of cellular communication network factor 4 (*CCN4*) was upregulated in PG-HSE grafts, compared to BTM-HSE (Figure 2D). In contrast, *Wnt5a*, a member of the non-canonical Wnt pathway, was overexpressed in both PG-HSE and BTM-HSE grafts, compared to other groups.

Integrin α6 is known to mediate attachment of basal keratinocytes to the basement membrane at the dermal/epidermal junction. Here, the engineered skin grafts were analysed for *ITGA6* expression. We found that although both PG-HSE and BTM-HSE grafts showed a significant *ITGA6* increase, compared to autograft, its upregulation was greater in BTM-HSE (*p* ≤ 0.0001). This was similar to the increase in other integrin alpha subunits in BTM-HSE (Figure 2C). This suggests that an essential skin homeostasis dermal/epidermal interaction between basal keratinocytes and their extracellular matrix exists in engineered skin grafts. Interestingly, collagen IV (*COLIV*), a major component of basement membrane, was not upregulated in either PG-HSE or BTM-HSE, compared to native skin (Appendix A). Furthermore, the ability of both PG and single layer BTM foam (plus human plasma) to maintain basal keratinocytes was supported by detection of cytokeratin 5^+^ cells in vivo and in vitro (Figure 3B,D). Cytokeratin 5^+^ basal keratinocytes were also detected in autografts due to antibody cross-reactivity.

### 2.4. PG-HSE Grafts Enhance Neo-Dermis Formation by Influencing Collagen Synthesis and Degradation

Collagen *COL1A1* (*p* ≤ 0.0001), *COL1A2* (*p* ≤ 0.0001), *COL3A1* (*p* ≤ 0.0001), *COL5A1* (*p* ≤ 0.0001), *COL5A2* (*p* ≤ 0.0001), and *COL5A3* (*p* ≤ 0.0001) mRNA expression was upregulated in PG-HSE, compared to BTM-HSE and autografts (Figure 4A), whereas matrix metalloproteinase, *MMP-1* (*p* ≤ 0.0001) and *MMP-9* (*p* ≤ 0.0001), responsible for collagen breakdown were overexpressed in BTM-HSE, compared to PG-HSE and autografts (Figure 4B).

Other remodelling enzymes, including cathepsin V (*CTSV*) and contactin 1 (*F3*), were also overexpressed in BTM-HSE, compared to other groups. Similarly, plasminogen activator, urokinase (*PLAU)* and plasminogen activator, urokinase receptor (*PLAUR*) were overexpressed in BTM-HSE (*p* ≤ 0.0001). Whereas an inhibitor of plasminogen activator inhibitor-1 (PAI-1; serine protease inhibitor clade E member 1 (*SERPINE1*)) was downregulated in both BTM-HSE and PG-HSE grafts, compared to autografts and ungrafted wounds (Figure 4B), cathepsin K (*CTSK*), however, was expressed at a higher level in PG-HSE grafts, compared to BTM-HSE grafts (*p* ≤ 0.0001). In addition, several well-known wound healing markers were unchanged both in PG-HSE and BTM-HSE grafts along with autografts (Appendix A). Overall, this molecular analysis predicts higher net collagen deposition in PG-HSE grafts, compared to BTM-HSE grafts (Appendix A). Trichome Masson staining of the grafts revealed fresh deposition of collagen within the grafts (Figure 4C,D). Although collagen levels did not reach the same levels as in the autografts, there was significantly more collagen in PG-HSE, within the grafts (*p* ≤ 0.05) and underneath the grafts in the wound bed (*p* ≤ 0.01), compared to BTM-HSE grafts. Picrosirius red staining identified higher type I collagen or COL I (red and yellow fibres, *p* ≤ 0.01) and type III collagen or COL III (green fibres, *p* ≤ 0.05) levels in PG-HSE grafts, compared to BTM-HSE grafts (Figure 5). Collagen deposited within the PG-HSE and BTM-HSE grafts had a mature basket-weave structure, similar to that seen in the autograft, whereas the collagen fibres deposited underneath the PG-HSE and BTM-HSE grafts had mostly a parallel appearance, indicating a lower level of maturity.

### 2.5. PG-HSE Grafting Restores Skin Barrier Function

A dermal phase metre (DPM) was used to provide a relative measure of the retained water content of the skin as a function of the skin’s dielectric value. The neo-epidermis restored a barrier against water loss and microbial invasion of wounds. Here, skin surface moisture was measured in PG-HSE and BTM-HSE grafts before (D0) and after (D14) grafting against autografts as an indication of skin barrier restoration. Despite a trend of higher water loss in BTM-HSE grafts, both PG-HSE and BTM-HSE formed a skin barrier within 14 days post-grafting, (Figure 6A).

### 2.6. PG-HSE Carries Intrinsic Anti-Microbial Properties That May Influence Graft Survival

A disc diffusion test was employed to study the intrinsic antimicrobial activity of the PG. Discs of the PG, BTM foam, and BTM foam (plus plasma) were placed on an Escherichia coli lawn and the zone of inhibition was measured. BTM foam (plus plasma) had some antimicrobial activity, mediated by plasma, but the platelet-derived hydrogel showed a significantly larger zone of inhibition, compared to both BTM foam and BTM foam (plus plasma) (*p* ≤ 0.0001) (Figure 6B).

## 3. Discussion

This study provides a mouse model for full thickness wound repair by PG-HSE and BTM-HSE engineered skin grafts. It sheds light on the molecular mediators that control this process. Involucrin detection in PG-HSE grafts showed a near normal differentiation pattern in the neo-epidermis in athymic nude mice. PG-HSE grafts also showed enhanced collagen synthesis and deposition, with a basket-weave structure in the neo-dermis, compared to a parallel alignment seen in the BTM-HSE grafts. This progress towards wound repair in PG-HSE grafts coincided with reduced inflammation in the wounds, compared to wounds grafted with BTM-HSE. The expression of inflammatory markers *CXCL-1*, *CXCL-2*, *IL-1β*, and *IL-6* were downregulated in the wounds grafted with PG-HSE, progressing the wound repair into the proliferation phase. The expression of *PTGS2*, which plays a central role in skin inflammation, was also lower in PG-HSE compared to BTM-HSE grafts. This was consistent with our earlier observation of high levels of COX2 and other inflammatory markers in wounds grafted with acellular BTM (with a seal) scaffold [10]. High levels of inflammation may have long-term consequences including scarring and fibrosis. In mice, COX2 was significantly lower in scarless healed wounds compared to wounds healed with scarring [21]. Long-term studies are needed to determine the effect of high inflammation in BTM-HSE grafts.

Trichrome Masson and Picrosirius red stains were employed to examine the structure of freshly deposited collagen in the engineered grafts. Although some collagen deposition was detected in engineered skin in vitro before grafting, there was significant collagen accumulation in the neo-dermis two weeks post-grafting. Both PG-HSE and BTM-HSE grafts showed the presence of mature basket-weave collagen within the grafts. In addition, parallel collagen fibres were abundant in the wound bed, deep to PG-HSE grafts, suggesting collagen synthesis stimulation in the PG-HSE graft wound bed. Upregulation of *COL1A1*, *COL1A2*, *COL3A1*, *COL5A1*, *COL5A2*, and *COL5A3* transcripts in PG-HSE grafts supported the above observation.

Although there was a significantly lower amount of fresh collagen deposited in BTM-HSE, compared to PG-HSE grafts, we found remodelling enzymes *MMP-1* and *MMP-9* were overexpressed in BTM-HSE grafts, compared to PG-HSE at the transcriptional level. This supported our earlier observation of MMP-9 upregulation in acellular BTM (with a seal) grafts [10]. Importantly, grafting a wound with epidermis has a modulatory effect on dermal fibroblasts and controls collagen synthesis, degradation, and remodelling. For example, a secretable form of stratifin from keratinocytes has a dual effect on fibroblasts during spontaneous wound repair. It downregulates collagen synthesis in fibroblasts, while promoting collagen degradation by enhancing expression of matrix metalloproteinases MMP-1, MMP-3, MMP-8, MMP-10, and MTP-MMP (MMP-24) [22,23]. It would be interesting to study whether stratifin affects collagen metabolism in PG-HSE and BTM-HSE grafts. The mechanism of MMP enzyme activation in BTM-HSE requires further investigation.

At two weeks post-grafting, autografts showed a low level of inflammation, along with collagen synthesis and remodelling. Autografts contain native resident fibroblasts within native extracellular matrix and, therefore, are not in an activated state. Here, we also showed upregulation of *SERPINE1* in ungrafted wounds. This finding agrees with reports of serine proteases such as urokinase plasminogen activator, tissue-type plasminogen activator (uPA/tPA), and their major physiological inhibitor, SERPINE1, being upregulated in several cell types during spontaneous injury repair [24]. This pathway was downregulated post-autografting, and even more effectively downregulated post-engineered skin grafting.

Ideally, a biomaterial scaffold in engineered skin would trigger regeneration rather than repair (or scar formation), replacing the normal skin tissue. The cell–scaffold interaction is one of the most important factors that control wound repair outcome post-grafting. Biomaterials modulate extracellular signalling with a profound impact on cellular activities. The biomaterials’ physical and biochemical composition instruct cell adhesion, and hence cellular fate. It is increasingly recognised that cell adhesion, mediated by integrin receptors, is strongly influenced by surface electrical charge, topography, and roughness [25,26]. Although, this study showed lower integrin expression in PG-HSE, compared to BTM-HSE grafts, the functional significance of this is yet to be determined. Conversely, the beneficial effect of non-autologous platelet-rich material in wound healing has attracted a lot of interest in recent years and, to our knowledge, this is the first study to show its effectiveness in engineering a permanent graft and wound closure in an animal model [27].

We have seen an upregulation of *TGF-β1*, a master regulator of fibrosis, in BTM-HSE grafts in this study and, previously, in BTM-alone grafts [28]. Plasminogen activator (*urokinase*) and urokinase receptor, upstream from TGF-β1 [29], were also highly expressed in the BTM-HSE graft. Similarly, *STAT3*, downstream from TGF-β1, was also overexpressed in the BTM-HSE graft. The functional consequence of high inflammation and TGF-β1 pathway activation in BTM-HSE grafts is not clear at this point. Similarly, TGF-β1 along with PDGF are enriched in platelet-derived hydrogel [17]. PDGF and TGF-β1 are considered pro-healing factors during the early phases of wound healing [30]. They attract pericytes and smooth muscle cells to the newly formed blood capillaries, which stabilise vessel walls in the granulation tissue [31]. However, high levels of TGF-β1 at later phases of wound healing are associated with fibrosis and scarring [32]. High TGF-β1 in the hydrogel may stimulate fibroblast to myofibroblast conversion in the early days and, therefore, promotes skin repair [33]. High concentrations of PDGF may also have a modulatory effect on scarring by increasing the expression of MMPs [34]. Moreover, insulin-like growth factor 1 (IGF-1), which is present in platelet-derived hydrogel, is enriched in amniotic fluid, providing a microenvironment for scarless foetal wound healing. It is important to note that wound repair is a complex process, and each particular wound repair marker could play a different, sometimes opposing, role, depending on its temporal expression during wound repair.

We investigated the amount of IL-10 in the graft microenvironment. IL-10 is known to mitigate fibrosis in other organs, including the heart, lung, kidney, liver, and intestine, and acts as a potent anti-inflammatory factor [35]. However, neither the *IL-10* transcript nor IL-10 protein were abundantly present in the engineered skin grafts. A limitation of this study is that it is short-term, focused on studying wound closure, and not showing the long-term effect of the PG-HSE or BTM-HSE grafts on scarring. Longer studies are needed to understand the scar outcome of both PG-HSE and BTM-HSE grafts.

One of the biggest challenges in engineered skin grafting is its susceptibility to infection. Here, we have presented some evidence that the PG-HSE graft may carry intrinsic antimicrobial characteristics that may support its survival. Our finding confirmed earlier reports of release of bacterial inhibitors, such as platelet factor 4 (PF-4), RANTES (CCL5), and connective tissue-activating peptide 3 (CTAP-3) from activated platelets with a direct mode of action, independent of immune cells [36]. Here, PG inhibitory action was only shown against E coli. It requires further investigation against clinically relevant bacterial strains. A meta-analysis identified that platelet rich plasma (PRP) improves skin graft take, but the effect of PRP against bacteria is not uniform and, therefore, it does not provide a treatment option for all bacterial infections [37,38]. A fundamental role of the skin is to create a barrier against heat, restrict fluid loss, and block microbial invasion. We have shown that the skin barrier was restored by both PG-HSE and BTM-HSE grafts in a full thickness wound mouse model. The potential inherent antimicrobial activity of the PG-HSE graft may improve barrier formation in contaminated wounds.

In conclusion, we have shown that the PG-HSE graft is as effective as an autograft and BTM-HSE graft in closing full thickness wounds in an athymic nude mouse model. It remains to be tested against autografts and BTM-HSE grafts for long-term scarring outcome.

## 4. Materials and Methods

### 4.1. Access to Human-Derived Material

Discarded skin tissue from adult donors (age 18–55) was obtained during elective breast reduction or abdominoplasty surgery after obtaining informed consent. This research project was approved by the Alfred Health Human Research Committee, Melbourne, Australia (Ethics Approval # 269/17). Donation of pooled human platelet precipitate and lysate was also approved by Australian Red Cross Lifeblood (Approval Number C Loh 18,072,014).

### 4.2. Isolation and Expansion of Primary Adult Fibroblasts and Keratinocytes

Adult fibroblasts were isolated from fresh skin, as described previously, with some modification [39]. Briefly, skin samples were digested in Dispase II (4 mg/mL in phosphate-buffered saline or PBS, Life Technologies, Carlsbad, CA, USA) overnight to separate the epidermis. The dermis was digested in Collagenase I (2.2 mg/mL, Life Technologies) for 45 min at 37 °C with constant agitation. IsFolated fibroblasts were expanded in low glucose Dulbecco’s Modified Eagle Medium (DMEM) with foetal bovine serum (4%, Sigma-Aldrich, St. Louis, MO, USA), hydrocortisone (0.5 µg/mL, Millipore, Burlington, MA, USA), insulin (50 IU/mL, Novo Nordisk, Bagsvaerd, Denmark), EGF (10 ng/mL, R&D Systems, Minneapolis, MN, USA), and gentamicin (50 µg/mL, Life Technologies). Adult keratinocytes were isolated from the epidermis, as described previously [10]. Briefly, epidermal sheets were digested with trypsin (0.25%, Life technologies) at 37 °C for 5–7 min to release the basal keratinocytes. Keratinocytes were expanded on irradiated 3T3-J2 feeder cells (Rheinwald’s laboratory, Boston, MA, USA) in cFAD (DMEM and Ham’s F12 (Life Technologies, 3:1 ratio), 4 mM L-Glutamine, 0.18 mM adenine (Merck, Darmstadt, Germany), 0.4 µg/mL hydrocortisone (Merck), 5 µg/mL insulin (Novo Nordisk), 2 × 10^9^ M 3,3′,5-triiodo-L-thyronine sodium salt (Sigma-Aldrich), 5 µg/mL transferrin (Sigma-Aldrich), 50 µg/mL gentamicin, 0.2 µg/mL isoproterenol (Hospira, Lake Forest, IL, USA), 10 ng/mL EGF (R&D Systems), and 10% foetal bovine serum (FBS), (Cytiva, Marlborough, MA, USA).

### 4.3. Fibroblast Proliferation Assay

Dermal fibroblast proliferation in a monolayer was measured using alamarBlue^®^ (Bio-Rad Laboratories, Inc., Hercules, CA, USA). Triplicate seedings of cells were left to attach in a 24-well plate. Fresh medium containing 10% alamarBlue was added and plates were incubated at 37 °C for 2 h. Finally, 100 µL aliquots from each well were transferred to a 96-well plate. Fluorescence measurements at 590 nm emission were collected using a FLUOstar OPTIMA plate reader (BMG Labtech, Ortenberg, Germany). Manual cell counting was performed using a hemocytometer under a brightfield microscope. Briefly, cells were diluted in trypan blue to achieve a countable concentration and introduced into the hemocytometer chamber. Total viable cell numbers were estimated, considering dilution factor and volume.

### 4.4. PG-HSE and BTM-HSE Construction

Adult dermal fibroblasts and basal keratinocytes were isolated and expanded as described previously [18]. A PG scaffold was constructed with some modifications [18]. Briefly, 2.4 IU/mL thrombin was added to the platelet-derived precipitate to form a hydrogel. A single layer BTM foam (plus plasma) scaffold was constructed, with some modifications [10]. Human plasma (60 mg/mL) was gelled in single layer BTM foam (no seal) for 30 min at 37 °C to fill pores in the scaffold. Both scaffolds (i.e., PG and single layer BTM foam (plus plasma)) were seeded with adult dermal fibroblasts (0.25 × 10^6^/cm^2^) on top of the scaffold, 24 h post-gelation. This was followed by seeding of basal keratinocytes (0. 5 × 10^6^/cm^2^). HSEs were matured in UCDM1 maturation medium consisting of low glucose DMEM (Life Technologies) 1:1 with Ham’s F12 medium (Life Technologies) plus 4.5 mM L-glutamine (Sigma-Aldrich), 1.5 mM L-serine (Sigma-Aldrich), 1 mM CaCl_2_ (Sigma-Aldrich), 1.8 mM adenine (Calbiochem, San Diego, CA, USA), 5 µM O-phosphorylethanolamine (Sigma-Aldrich), gentamicin (50 µg/mL, Life Technologies), 1 mM strontium chloride (Sigma-Aldrich), 1 x ITS (Sigma-Aldrich), linoleic acid (10 µg/mL, Sigma-Aldrich), 0.1 mM ascorbic acid (Sigma-Aldrich), 20 pM triiodothyronine (Sigma-Aldrich), hydrocortisone (0.5 µg/mL, Merck), KGF (5 ng/mL, R&D Systems), EGF (10 ng/mL, R&D Systems), and FGF-2 (1 ng/mL, R&D Systems) supplemented with 0.3% FBS for 5 days (PG-HSE) or 12 days (BTM-HSE) [20].

### 4.5. Mouse Surgery

The protocol was reviewed and approved by the Alfred Research Alliance Animal Ethics Committee (Ethics approval # E/1920/2019/A) to conform with the NIH Guide for Care and Use of Laboratory Animals. Surgery was performed as previously described, with some modifications [17]. Briefly, athymic nude mice aged 8–10 weeks were anaesthetised with isoflurane (2 L/min) and a full thickness surgical wound created on the dorsal side by excising a circular, 1.5 cm in diameter, section of skin. A custom-made silicone (Life Technologies) ring was attached to the outer edges of the wound using Histoacryl skin glue (B. Braun Surgical, Melsungen, Germany) and sutured (monofilament nylon, B. Braun Surgical) with 6 to 8 simple interrupted stitches along the outer circumference of the ring. The PG-HSE or BTM-HSE graft was placed on the wound, dressed, and sealed with SurfaSoft^®^ (Tauren, Shandong, China), Tegaderm™ (3M Health Care, St. Paul, MN, USA) and Nexcare™ (3M Health Care) for the duration of the experiment. Wound dressings were changed every 5 days, mice were euthanised and grafted wounds were analysed after 14 days.

### 4.6. Histological Analysis

Grafted wound sections were stained with haematoxylin and eosin, Masson’s trichrome solution according to standard protocols. Briefly, sections were dewaxed with 3 changes in xylene and rehydrated using 100%, 95%, and 70% ethanol, followed by 25 min incubation at 65 °C. Slides were rinsed with distilled water for 1 min and placed in Bouin’s fixative for 1 h at 65 °C. Sections were stained with Celestin Blue (Cat. No. 029-ADCB-25GM, Pathtech, Preston, Australia) for 5 min and Weigert’s hematoxylin (Hematoxylin: Cat. No. 75290, Hurst Scientific, Forestdale, Australia and Ferric chloride: Cat. No. 157740-1KG, Merck) for 30 min, followed by differentiation in acid alcohol. Sections were washed and stained with Biebrich scarlet (Cat. No. SKU BS-25G, Hurst Scientific) plus acid fuchsin (Cat. No. 227901000, Thermo Fisher Scientific, Waltham, MA, USA) for 5 min. Sections were incubated in 5% Tungstophosphoric Acid (Cat. No. P4006, Merck) for 5 min, followed by a water rinse. Sections were counterstained in aniline blue (Cat. No. C066, ProSciTech, Kirwan, Australia), dehydrated in 95% ethanol, and cleared with 3 changes in xylene. Sections were mounted using an automated Leica coverslipper. Stained sections were imaged using an Olympus BX51 microscope (20×/0.75 objective) and Olympus colour DP70 camera. Fresh collagen deposition was quantified using Image J software abcam 1.52d (NIH). Briefly, a region of interest (ROI) was manually drawn. The threshold on ROI manager was adjusted to encompass all stained area. This allowed the generation of integrated density readings for statistical analysis.

Tissue slides were stained with Picrosirius red (Abcam, Cambridge, UK) and imaged using an Olympus VS200 slide scanner under polarised light. An ImageJ macro (PR-BRF Analyser v2.20, CCM Lab, Darlinghurst, Australia) was used to measure areas occupied by each colour [40]. Images were converted to 8-bit and colour ranges were assigned to collagen fibre types as follows: red (0–13, 240–255, long/mature), yellow (14–43, mixed/intermediate), and green (44–140, thin) to reflect controls including collagen I, human native skin, and mouse native skin. Five randomly selected regions of 360 × 240 µm within the wound bed and beneath the graft were analysed for collagen composition.

### 4.7. Immunofluorescent Staining

Slides were blocked as above and incubated with primary antibodies: rabbit anti-human involucrin (1:100, Cat. No. ab234403, Abcam), rabbit anti-human Ku80 (1:50, Cat. No. 214745, Biorbyt, Durham, NC, USA), or rabbit anti-human CK5 (1:250, Cat. No. 905504, BioLegend, San Diego, CA, USA ) overnight. Slides were washed and incubated with donkey anti-rabbit IgG Alexa Fluor 647-conjugated antibody (1:400, Cat. No. A31573, Invitrogen, Waltham, MA, USA). All images were captured using a Nikon TiE inverted microscope (Tokyo, Japan) equipped with a Plan Apo (40×/0.95l objective] objective and an Andor Zyla 4.2 sCMOS camera. Quantitative image processing and analysis were performed using Nikon NIS-Elements Analysis software (version 5.5), incorporating the Clarify.ai feature to enhance image contrast and detail. Staining was quantitated for integrated density, as the sum of pixel value/signal intensity within a selected ROI or counted for total/positively stained cells. Thresholds were adjusted to minimise background using negative control staining.

### 4.8. Wound Contraction

Wound images of 5 mice per group were taken with a scale on days 0, 5, 10, and 14. Wound area (cm^2^) was measured using FIJI (Image J) software (NIH). Briefly, a scale bar was set using the known scale in an opened image. A perimeter was drawn to select the wound area as the ROI using polygon selections. Wound area (cm^2^) was recorded for statistical analysis.

### 4.9. Skin Barrier Function

Skin barrier function or relative hydration of the skin was measured using a dermal phase metre, model DPM 9003 (NOVA Technology Corporation, Shenzhen, China), as per the manufacturer’s instructions. Skin surface impedance was measured to determine electroconductivity of the test sites.

### 4.10. RT^2^ Profiler PCR Array

Skin grafts were trimmed on the edge, homogenised with an IKA Ultra-Turrax T-25 disperser (Janke and Kunkel, Staufen, Germany), and processed for RNA extraction using the RNeasy mini kit (Qiagen, Hilden, Germany). Eluted RNA received 10 U RNasin^®^ Plus RNase inhibitor (Promega, Madison, WI, USA) and was quantified on a Quantus™ Fluorometer using a QuantiFluor^®^ RNA System kit (Promega). Human Wound Healing RT^2^ PCR Profiler array (Cat. No. 330231, Qiagen) was set up for test and control samples. This array identifies both human and mouse genes due to their high homology level.

### 4.11. Disc Diffusion Antibacterial Test

PG, single layer BTM foam, single layer BTM foam plus plasma gel, and a fibrin gel were prepared. Gentamicin (Thermo Fisher Scientific) and kanamycin (Gibco, Waltham, MA, USA) were added to PBS at 30 µg, 15 µg, or 5 µg. These were loaded over sterile filter paper discs (6 mm in diameter) as positive controls. A sterile filter paper disc (6 mm) loaded with antibiotic diluent (sterile PBS) was used as the negative control. Mueller Hinton agar plates were inoculated with E-coli (10^5^ CFU/mL) using 100 ul LB agar broth. Discs were loaded on the top of Mueller Hinton agar plates and incubated at 37 °C for 16–18 h. The inhibition zone was traced and the diameter (mm) measured using FIJI software (NIH). The inhibition zone diameter was considered as an indication of the level of antibacterial activity.

### 4.12. Statistical Analysis

Data were analysed using ordinary one-way ANOVA or two-way ANOVA and Bonferroni’s post hoc test. *p*-values of less than 0.05 were sufficient to reject the null hypothesis for all analyses.

## Figures and Tables

**Figure 1 ijms-26-09988-f001:**
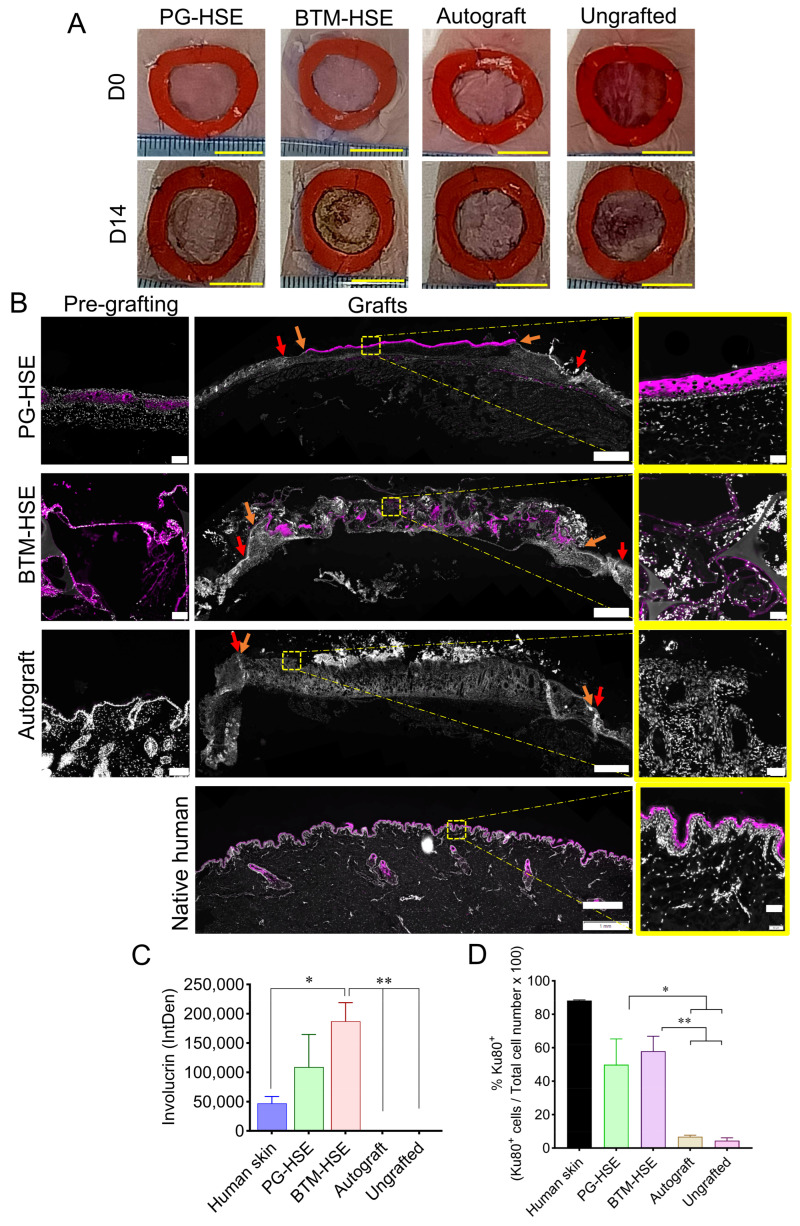
**Full thickness wound repair by engineered skin grafting in athymic nude mice**. (**A**) Representative images of the wounds grafted with PG-HSE, BTM-HSE, or native skin autografts on day of grafting (day 0) and post-grafting (day 14). (**B**) Representative microscopic images of day 14 grafts analysed by immunofluorescence, using a human specific antibody against involucrin. PG-HSE pre-grafting (scale 100 µm); PG-HSE graft (scale 1 mm); BTM-HSE pre-grafting (scale 100 µm); BTM-HSE graft (scale 1 mm); autograft pre-grafting (scale 100 µm; autograft (scale 1 mm); and human native skin (scale 1 mm). Zoomed images scale bar = 50 µm. Orange arrows indicate the graft edge 2 weeks post-grafting and red arrows indicate the initial wound edge. (**C**) Presence of human involucrin in grafts is presented as integrated density (IntDen), that is, sum of pixel value/signal intensity within a selected region of interest calculated using FIJI software version 1.54p. (**D**) Human-specific Ku80 marker was detected by immunofluorescence on day 14 post-grafting and % Ku80 positive cells were counted using Nikon NIS-Elements Analysis software (version 5.5). Data were analysed using ordinary one-way ANOVA. Values in (**C**,**D**) represent mean values +/− SEM in each group (* = *p* ≤ 0.05, ** = *p* ≤ 0.01, n = 4–5 per group). PG-HSE, platelet-derived human skin equivalent; BTM-HSE, NovoSorb biodegradable temporising matrix human skin equivalent; autograft, and autologous full thickness skin graft.

**Figure 2 ijms-26-09988-f002:**
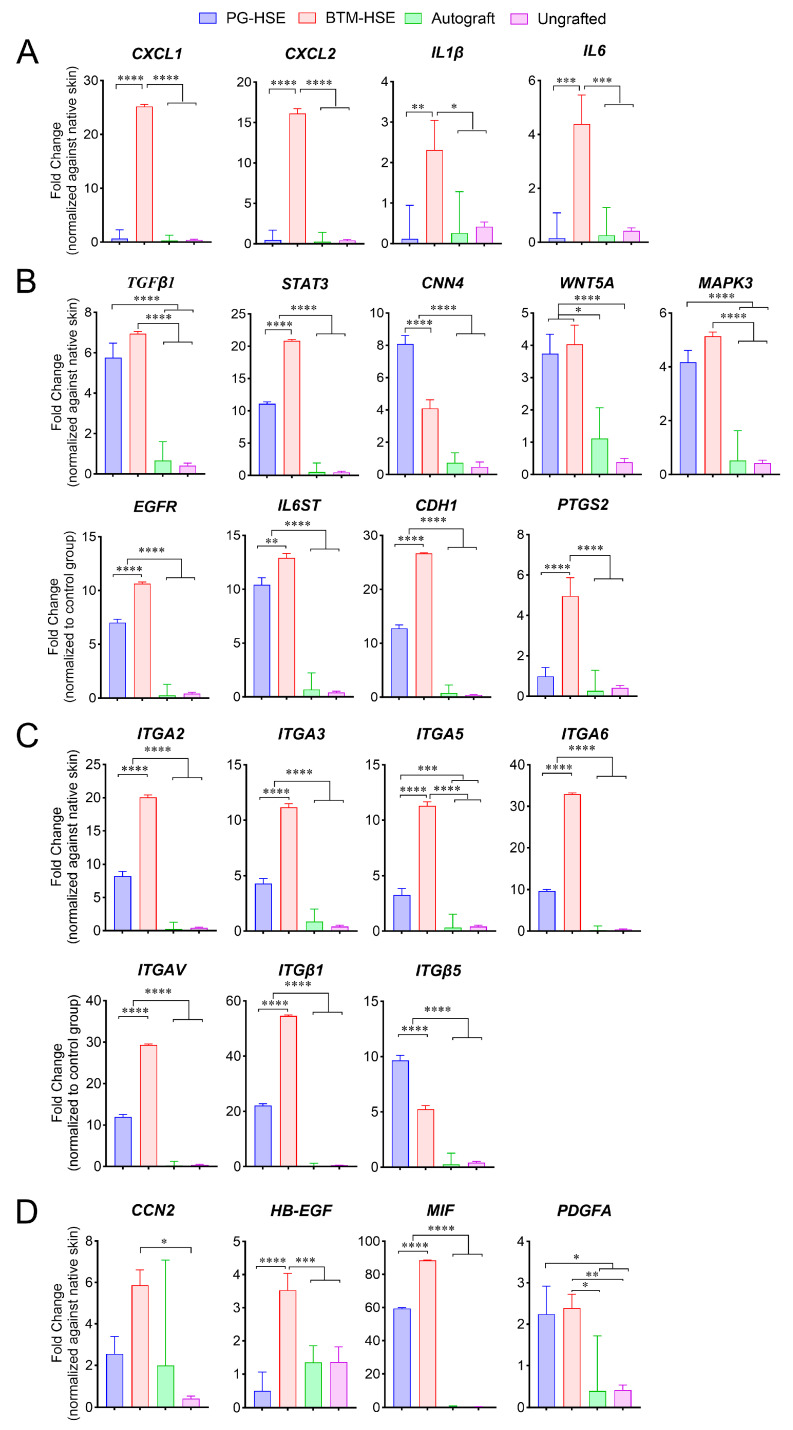
**Effect of engineered skin grafts on transcription of wound healing markers.** (**A**) Inflammatory cytokines and chemokines, (**B**) cell surface receptor signal transduction, (**C**) cell adhesion molecules, and (**D**) growth factor markers mRNA expression were measured using wound healing RT^2^ profiler PCR array. Data were analysed using QIAGEN GeneGlobe Data Analysis Center web resource. Statistical analysis was performed using ordinary one-way ANOVA. Values represent mean +/− SEM in each group (* = *p* ≤ 0.05, ** = *p* ≤ 0.01, *** = *p* ≤ 0.001, **** = *p* ≤ 0.0001, n = 4–5 mice per group). PG-HSE, platelet-derived human skin equivalent; BTM-HSE, NovoSorb Biodegradable temporising matrix human skin equivalent; and autograft, autologous full thickness skin graft.

**Figure 3 ijms-26-09988-f003:**
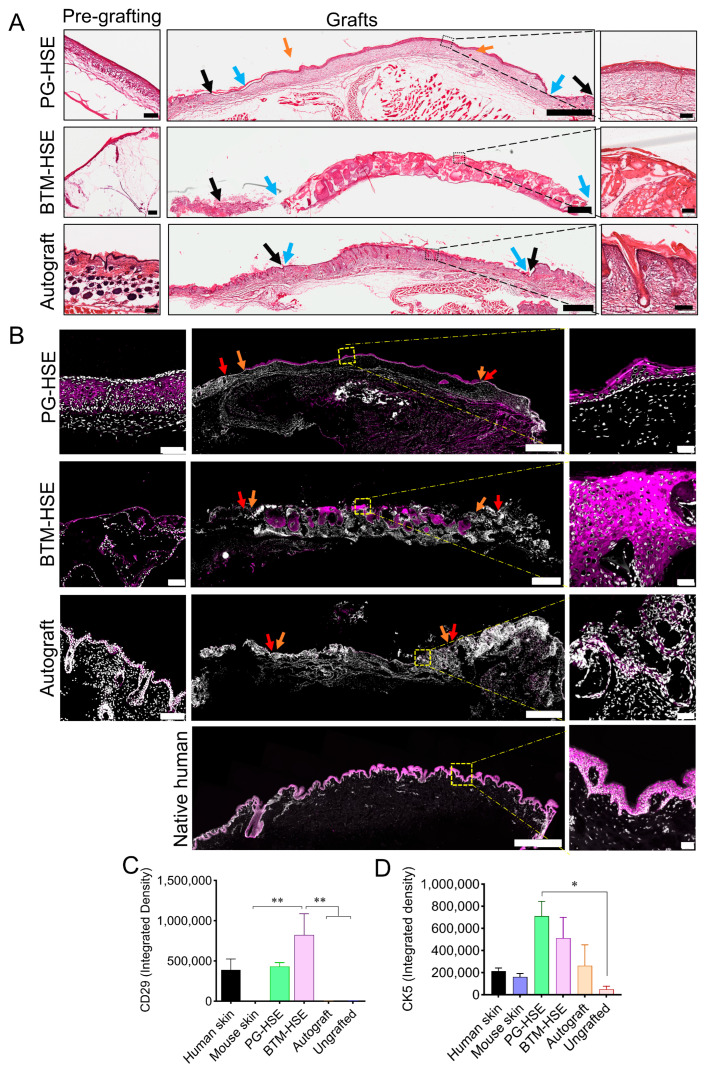
**Presence of basal keratinocytes in engineered skin grafts.** (**A**) Grafts were analysed 2 weeks post-grafting by haematoxylin and eosin staining: PG-HSE before grafting (scale bar = 100 µm); PG-HSE graft (scale bar = 1 mm); BTM-HSE before grafting (scale bar = 100 µm); BTM-HSE graft (scale bar = 1 mm); mouse native skin (scale bar = 100 µm); and autograft (scale bar = 1 mm). Zoomed images show scale bar = 100 µm. Blue arrows indicate graft edge two weeks post-grafting and black arrows show the original wound edge. (**B**) Representative data showing detection of CK5 in PG-HSE pre-grafting (scale bar = 100 µm); PG-HSE graft (scale bar = 1 mm); BTM-HSE pre-grafting (scale bar = 100 µm); BTM-HSE graft (scale bar = 1 mm); autograft pre-grafting (scale bar = 100 µm), autograft graft (scale bar = 1 mm); and human native skin (scale bar = 1mm) using immunofluorescence. Zoomed images scale bar = 50 µm. Orange arrows indicate the graft edge and red arrows show the original wound edge. Grafts were also analysed by immunofluorescence using (**C**) integrin β1/CD29-specific antibody and (**D**) CK5-specific antibody. Both CD29 and CK5 were quantified by calculating integrated density using FIJI software. Data were analysed using ordinary one-way ANOVA. Mean values +/− SEM in each group (* = *p* ≤ 0.05, ** = *p* ≤ 0.01, n = 4–5 per group) are presented. PG-HSE, platelet-derived human skin equivalent; BTM-HSE, NovoSorb biodegradable temporising matrix human skin equivalent; and autograft, autologous full thickness skin graft.

**Figure 4 ijms-26-09988-f004:**
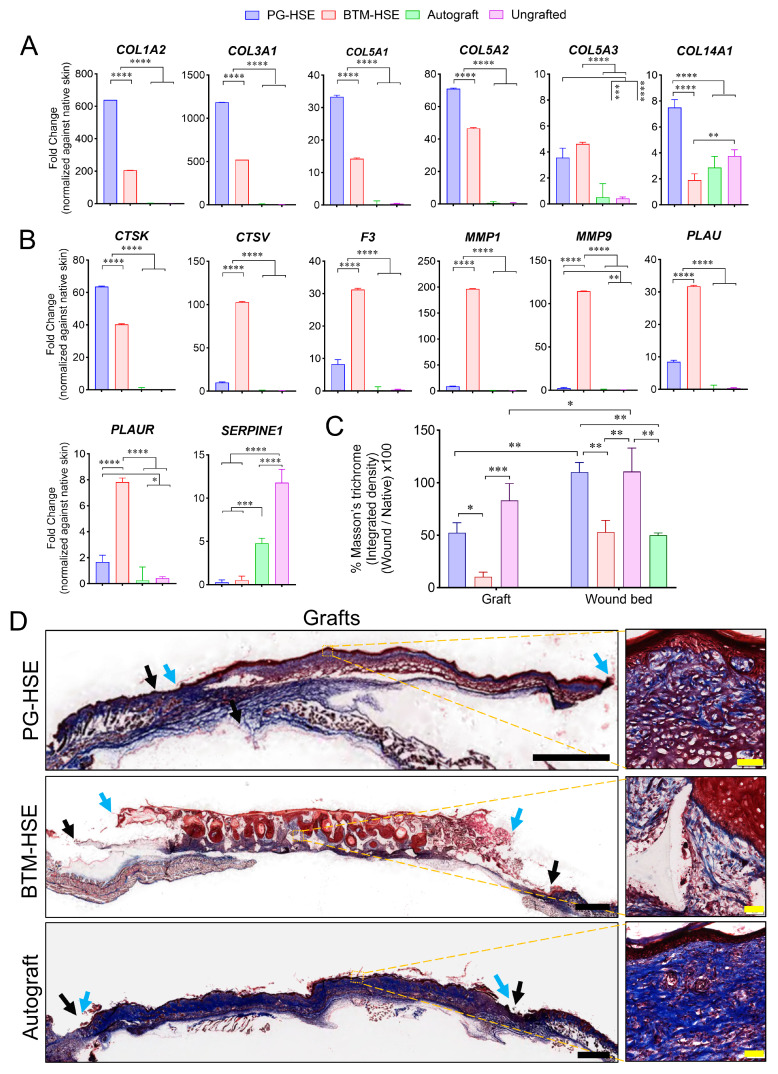
**Neo-dermis in engineered skin grafts.** (**A**) Collagen and (**B**) ECM remodelling enzymes expression were measured in grafts at the transcriptional level. (**C**) Collagen was detected in grafts and wound bed (below grafts) using Masson’s trichrome (MT) staining. (**C**) %MT: the percentage of integrated density of MT in grafts was normalised against mouse native skin. (**D**) Representative microscopic images of MT staining in PG-HSE graft (scale bar = 1 mm); BTM-HSE graft (scale bar = 1 mm); and autograft (scale bar = 1 mm). Zoomed images scale bar = 50 µm. Blue arrows indicate graft edge and black arrow shows the original wound edge. Data in (**A**,**B**) were analysed using ordinary one-way ANOVA and (**C**) was analysed using two-way ANOVA then Bonferroni’s post hoc test. Values represent mean values +/− SEM in each group (* = *p* ≤ 0.05, ** = *p* ≤ 0.01, *** = *p* ≤ 0.001, **** = *p* ≤ 0.0001, n= 4–5 per group). PG-HSE, platelet-derived human skin equivalent; BTM-HSE, NovoSorb biodegradable temporising matrix human skin equivalent; and autograft, autologous full thickness skin graft.

**Figure 5 ijms-26-09988-f005:**
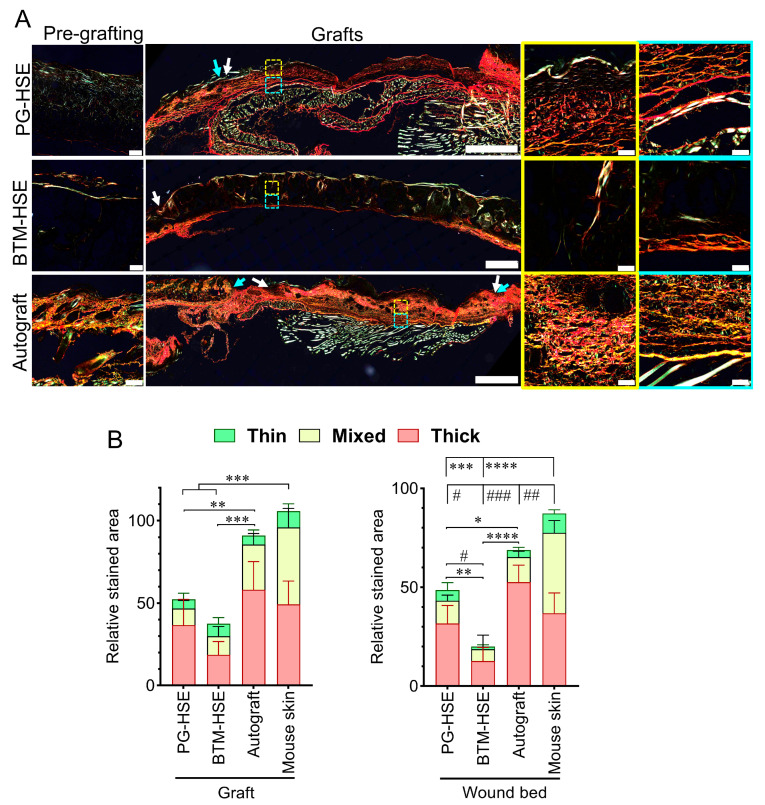
**Collagen composition and arrangement in engineered grafts using Picrosirius red stain.** (**A**) Representative microscopic images of PG-HSE pre-grafting (scale bar = 100 µm); PG-HSE graft on day 14 (scale bar =1 mm); BTM-HSE pre-grafting (scale bar = 100 µm); BTM-HSE graft on day 14 (scale bar = 1 mm), autograft pre-grafting (scale bar = 100 µm), and autograft on day 14 (scale bar = 1 mm). The yellow outlined zoomed images represent the graft area and the blue outlined zoom images represent the wound bed area (scale bar = 50 µm). White arrows indicate graft edge and blue arrows show the original wound edge. (**B**) Quantification of Col I (yellow and red) and Col III (green) were measured against the whole wound area using FIJI software on day 14 post-grafting. Data were analysed using two-way ANOVA then Bonferroni’s post hoc test. Values represent mean values +/− SD in each group, * represents statistical significance of Col I (* = *p* ≤ 0.05, ** = *p* ≤ 0.01, *** = *p* ≤ 0.001, **** = *p* ≤ 0.0001; while # represents statistical significance of Col III (# = *p* ≤ 0.05, ## = *p* ≤ 0.01, ### = *p* ≤ 0.001, n = 4–5 mice per group). PG-HSE, platelet-derived human skin equivalent; BTM-HSE, NovoSorb biodegradable temporising matrix human skin equivalent; and autograft, autologous full thickness skin graft.

**Figure 6 ijms-26-09988-f006:**
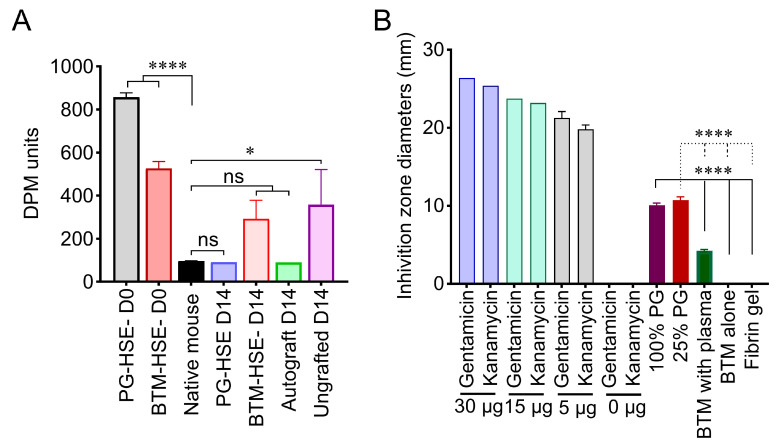
**The functional outcome of engineered skin grafting.** (**A**) Barrier functions of PG-HSE pre-grafting (day 0); PG-HSE grafts (day 14); BTM-HSE pre-grafting (day 0); BTM-HSE grafts (day 14); autograft and ungrafted wounds were measured and analysed against mouse native skin. (**B**) Antimicrobial effects of 100% and 25% PG, BTM foam with or without plasma clot, and fibrin gel compared with standards gentamicin and kanamycin antibiotics. Statistical analysis was performed using ordinary one-way ANOVA. Values represent mean values +/− SEM in each group (∗ = *p* ≤ 0.05, ∗∗∗∗ = *p* ≤ 0.0001, n = 3–5 mice per group). PG-HSE, platelet-derived human skin equivalent; BTM-HSE, NovoSorb biodegradable temporising matrix human skin equivalent; autograft, autologous full thickness skin graft; and DPM, dermal phase metre.

## Data Availability

Data are contained within the article or Appendix A.

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
