# Peer review of "Bioengineered Skin from a Platelet-Derived Hydrogel Repairs Full Thickness Wounds in a Pre-Clinical Mouse Model"

_ijms, 2025, doi:10.3390/ijms26209988_

Round 1

Reviewer 1 Report

Comments and Suggestions for Authors

This manuscript presents the platelet-derived hydrogel for full thickness wounds repair. The topic is certain meaningful. The framework is clear and the results could support the authors' views. In summary, I recommend publishing this manuscript in “International Journal of Molecular Sciences”. However, there are still some problems that need to be improved before publication. And the following are some suggestions for revision.

  1. The introduction of the manuscript should be improved to present the purpose and significance of the manuscript.
  2. The titles of Results section should be revised to more concise.
  3. It is suggested to prepare manuscript according to the format requirements of the journal, like the linewidth of the Section 2.1 is different with other sections.
  4. It is better to cite more latest references.
  5. It is suggested to include photos of the wound at different time points to better illustrate the healing process.
  6. The figures (Fig. 1, 2, 3 and 4) in the manuscript should be recombined to make them more scientific and artistic.
  7. It is better to give more discussion on the mechanism of full thickness wound healing.

Author Response

This manuscript presents the platelet-derived hydrogel for full thickness wounds repair. The topic is certain meaningful. The framework is clear and the results could support the authors' views. In summary, I recommend publishing this manuscript in “International Journal of Molecular Sciences”. However, there are still some problems that need to be improved before publication. And the following are some suggestions for revision.  

Many thanks for your review and recommendations. Please find below the response to your specific comments.   

  1. The introduction of the manuscript should be improved to present the purpose and significance of the manuscript.

Response: Introduction has been improved to highlight the significance of this study. “

." Apart from the traditional autologous thin skin graft that is harvested from a donor site, engineered skin from expanded basal keratinocytes can provide a viable alternative.[3].

In order to provide an appropriate micro environment for basal keratinocytes, a wide range of biomaterials such as collagen-based matrices, hydrogels, silk, biodegradable polyurethane or their combinations have been tested as scaffolds over the last few decades, each with inherent advantages and disadvantages.

  1. The titles of Results section should be revised to more concise.

Response: The results subheadings have been revised.

  • Platelet-derived hydrogel is an effective scaffold for constructing a PG-HSE graft and closing wounds in a mouse model.
  • PG-HSE grafts show lower transcription levels of inflammatory markers, compared to BTM-HSE grafts.
  • Basal keratinocytes are sustained in both PG-HSE and BTM-HSE grafts

  1. It is suggested to prepare manuscript according to the format requirements of the journal, like the linewidth of the Section 2.1 is different with other sections.

Response: Corrected as suggested. Thank you.

  1. It is better to cite more latest references.

Response: Cited additional latest references.

  1. It is suggested to include photos of the wound at different time points to better illustrate the healing process.

Response: Thank you for your suggestion. We agree it would be better to add photos of grafts at the dressing change. This however requires a lot of space in the figure.

  1. The figures (Fig. 1, 2, 3 and 4) in the manuscript should be recombined to make them more scientific and artistic.

Response: Figures 1, 2, 3 and 4 have been recombined as recommended to improve their scientific and artistic illustration. Thank you.

  1. It is better to give more discussion on the mechanism of full thickness wound healing.

Response: Many thanks for your comment. Spontaneous wound healing mechanism is comprehensively discussed in many of our referred papers. Our focus is the wound repair by skin grafting which is not as well described.

NOTE: Thank you for your offer of language editing. This manuscript has been edited with a professional editor and does not require any further language editing.

Reviewer 2 Report

Comments and Suggestions for Authors

Author Response

Bioengineered Skin from a Platelet-Derived Hydrogel Repairs Full Thickness Wounds in a Pre-Clinical Mouse Model

Response: Many thanks for your review and recommendations. Please find below the response to your specific comments.   

Suggestion: minor revisions

  1. Please provide a simple diagram of skin equivalents and show the difference between ‘with seals’ and ‘without a seal’ for those in your reader-ship who may not be aware of it.

Response: Thank you for your suggestion. In this project, we used single layer BTM foam (without the seal) with plasma as matrix to manufacture BTM-HSE. This has been added to methods and an illustration has been added to the supplementary figure.

  1. Please provide a simple figure of the layers of the skin and clearly indicate the areas that you were trying to replicate in your work.

Response: Thank you for your comment. PG-HSE is composite and replicates both dermis and epidermis. This is added to the illustration above.

  1. For a lot of your bar graphs a number of reading are not visible as their values are less than 1. I wonder if presenting data in log scale might be better? Please feel free to ignore this, if no other reviewer comments on this.

Response: We changed font size for all the graphs and images to make them more visible. To represent all the bar graphs consistent and to make our story clearer we kept x-axis similar, especially for statistically non-significant bar graph.

  1. Please provide a simplified figure of the main 2-3 pathways that according to you and your current data is mainly involved in wound healing in skin. You may add this as a supplementary figure.

Response: This is added to the illustration above.

  1. Please expand on your limitation and future directions – assuming you are preparing to take this to clinical study. What al changes would you make then and how would you ensure reproducibility of your work to ensure safety and efficacy of you hydrogels?

Response: Thanks for your comment. As a primary cellular therapy (autologous human skin equivalent) there are challenges including batch to batch variation. Over the last few years we have developed a quality system and adapted semi-automatic manufacturing system to avoid those challenges especially batch to batch variation. Further we receive GMP grade pooled platelet precipitate from our collaborator, Australian Red Cross, which is tested both in in-vivo & in-vitro models and published earlier (ref 17 and 18) to ensure safety and efficacy.

  1. Please check the allowed number of self-citation with the journal, I think it is 15% and currently you are just over that.

Response: This manuscript is a continuation from our in-vitro development and analysis of PG-HSE. We have added latest references related to this study and self-citation is now within the 15% limit.

NOTE: This manuscript has been edited with a professional editor and does not require any further language editing.
